# Optimization of Zebrafish Larvae Sectioning for Mass Spectrometry Imaging

**DOI:** 10.3390/ph15101230

**Published:** 2022-10-07

**Authors:** Junhai Yang, Lauren Rendino, Steven Cassar, Wayne Buck, James Sawicki, Nari Talaty, David Wagner

**Affiliations:** AbbVie Inc., 1 Waukegan Rd., North Chicago, IL 60064, USA

**Keywords:** MALDI imaging mass spectrometry, zebrafish larvae, sectioning, embedding medium

## Abstract

The utility of zebrafish is becoming more frequent due to lower costs and high similarities to humans. Zebrafish larvae are attractive subjects for drug screening and drug metabolism research. However, obtaining good quality zebrafish larvae sections for batch samples at designated planes, angles, and locations for comparison purposes is a challenging task. We report here the optimization of fresh frozen zebrafish larvae sectioning for mass spectrometry imaging. We utilized the gelatin solutions that were created at two different temperatures (50 and 85 °C) as embedding media. Gelatin-50 (gelatin created under 50 °C, solid gel under room temperature) was used to make a larvae-shaped mold and gelatin-85 (gelatin created under 85 °C, liquid under room temperature) was used to embed the larvae. H&E staining of sections shows well-preserved morphology and minimal histological interference. More importantly, the position of the larvae was well controlled resulting in more consistent sectioning of the larvae.

## 1. Introduction

The principles of the 3Rs, Replacement, Reduction, and Refinement, are being increasingly incorporated into legislations, guidelines, and practice of animal experiments in order to safeguard animal welfare [1]. Among the many methods that lead the way towards a more humane situation in animal testing, the use of zebrafish larvae follows the principle of the 3Rs as required by national and international regulatory bodies. Zebrafish are not deemed animals until they are capable of independent feeding. The utility of zebrafish is becoming more frequent due to lower costs and high similarities to humans. Zebrafish embryos develop the major organ systems present in mammals including the cardiovascular, nervous, and digestive systems in 7 days [2]. Additionally, zebrafish larvae can absorb compounds through water [3] and their raising cost is low. All these features make zebrafish larvae attractive subjects for drug screening and drug metabolism research [4,5].

Mass spectrometry imaging can detect hundreds of molecules on the tissue surface directly and simultaneously without the need to use antibodies or labels [6]. However, obtaining good quality zebrafish larvae sections for batch samples at designated planes, angles, and locations for comparison purposes is a challenging task. Especially, as zebrafish larvae have a much smaller size (2–3 mm in length) than adult zebrafish (2–5 cm in length), embedding the larvae is necessary for sectioning. Since most of the embedding medium is not transparent when frozen, without depth control of the embedding, it is difficult to locate the larvae in the embedding block resulting in the low throughput of the sectioning process. Furthermore, the embedding medium is required to have a minimal mass spectrum background to avoid interference to the detection of compounds of interest. The preference for the embedding process is to be at room temperature or lower to reduce frozen artifacts and this requires the embedding medium to stay liquid under the said conditions.

Although the sectioning, embedding, and staining of formaldehyde-fixed zebrafish embryos using plastic resin was reported previously [7], the procedure that is suitable for fresh frozen zebrafish larvae for mass spectrometry imaging is lacking. Nelson et al. investigated 13 distinct embedding media formulations for adult zebrafish, and they found that the 10% gelatin and 5% CMC (carboxymethyl cellulose) mixture is the optimal embedding medium [8], however, this gelatin/CMC mixture only remains in liquid form at 37 °C or higher, which is not ideal for preserving the small molecule distribution in zebrafish larvae. Marta et al. developed 3D-printable tools that make the placement and positioning of the larvae much more reliable and consistent for improving embryo injection and screening [9], although, sectioning of the fresh frozen larvae was not utilized in their application. Mariana et al. embedded zebrafish larvae in 10% gelatin under 50 °C successfully [10]. However, our applications necessitate the sectioning of a large number of fish larvae, therefore, requiring higher throughput and improved control of the larvae embedding depth and positioning in the cutting block.

Here, in this application note, we report the optimization of fresh frozen zebrafish larvae sectioning for mass spectrometry imaging. We utilized the gelatin solutions that were created at two different temperatures (50 and 85 °C) as embedding media. Gelatin-50 (gelatin created under 50 °C, solid gel under room temperature) was used to make larvae-shaped molds and gelatin-85 (gelatin created under 85 °C, liquid under room temperature) was used to embed the larvae. During sectioning, the PTFE surface was used to assist section placement. Using these conditions, the larvae sections with minimum folding and defects were obtained. H&E (hematoxylin and eosin stain) staining of sections showed a well-preserved morphology and no histological interference from gelatin-85. More importantly, the positioning of larvae was well controlled resulting in many more reproducible sections.

## 2. Results and Discussion

### 2.1. Considerations of Embedding

It is important to control the embedding depth of the zebrafish larvae in the cutting block to have high-throughput sectioning. It is also critical to control the orientation and angles of the larvae body on the section plane to obtain sections reproducible at the location suitable for the interpretation of the pathology and biology. Furthermore, there are requirements particularly for mass spectrometry imaging of zebrafish larvae:Embedding medium has little interference and ion suppression towards detection.Embedding medium brings minimal interference for the interpretation of the stained sections.Sections will need to be preserved well and have no defects and freezing artifacts.

#### 2.1.1. Embedding Medium Selection Based on Minimal Mass Spectrum Background

During an MALDI-MSI analysis, both matrix and embedding media will introduce gas phase ions that potentially cause suppression and interference when detecting target compound(s), especially in the mass range of small molecule drugs (*m*/*z* 500–1500). Among the embedding media that was tested, gelatin-85 and the gelatin/CMC mixture provide a minimal mass spectral background signal (number of ions and intensity), hence, the smallest potential for interference in the detection of small molecules (150–1800 *m*/*z*). This is shown in Figure 1, where the full mass spectrum is divided into three mass ranges to facilitate comparisons across various embedding media. In addition, the scaling of the mass ranges is constant across all conditions to simplify the evaluation. HPVP/PMC and the common embedding media OCT display the highest number of intense ions while gelatin-85 and gelatin/CMC exhibit a much lower background as shown in Figure 1. The interference of OCT toward mass spectrometry detection has been previously reported [11,12]. More details on the background spectra are located in Appendix A.

#### 2.1.2. Embedding Medium Selection Based on the Embedding Properties of the Medium

Gelatin-85 is a liquid at room temperature, as a result, zebrafish larvae can be embedded at ambient conditions. This makes the adjustment of larvae body position an easy task when managing several samples. The 10% gelatin and 5% CMC mixture displays effective sectioning properties, low ion suppression, and appropriate preservation of histology of sections [8]. However, this mixture is a partially solid gel at room temperature and necessitates temperatures above 37 °C to be in a liquid state suitable for the embedding process. In addition, its high viscosity makes it time consuming for positioning the larvae, leading to poor execution during the embedding of a large number of samples. HPMC/PVP (hydroxypropyl)-methylcellulose and polyvinylpyrrolidone mixture) can produce high-quality sections and does not introduce interference to histological stains [13], but similar to the gelatin/CMC mixture, HPMC/PVP has a high viscosity and requires to be at least 37 °C during the embedding process.

Overall, based on the above investigation, the gelatin-85 solution was chosen as the best option for the embedding medium as it displays the minimal mass spectrometry interference and provides excellent embedding properties.

### 2.2. Embedding Process with the Zebrafish-Shaped Mold

#### 2.2.1. The 3D-Printed Zebrafish Larvae Mold for Controlled Embedding and Sectioning

To control the larvae placement in the cutting block and increase the sectioning throughput, the zebrafish were embedded in a custom gelatin mold as shown in Figure 2b. The gelatin mold was constructed from a 3D-printed casting mold based on the specifics that were reported previously [14] (Figure 2a). With this approach, each larva is placed in the well of a similar size enabling each larvae body to be oriented parallel to the cutting plane. Furthermore, the embedding depths of larvae were controlled (about 200 µm from the surface of the block), facilitating sectioning speed and reproducibility. This greatly reduced the randomness of the sectioning process (3 steps of trimming at 50 µm/step, followed by 4–5 steps at 12 µm/step). As a result, one larva embedded in a mold is sectioned in 20 min (25–30 sections of 12 µm thickness), which is much quicker than sectioning larvae not in the mold (60 min) as the major portion of the time was locating the larva and adjusting the sectioning plane for the random order.

#### 2.2.2. Advantages of Using Two Gelatin Solutions Prepared under 50 and 85 °C for Embedding

It is critical to use the same medium for making the mold and embedding. When using a gelatin/CMC mixture to embed the larvae in the gelatin-50 mold, we observed larvae sections (in gelatin/CMC) detaching from the mold location (gelatin) during the sectioning as shown in Figure 3a. It is possible that this was caused by the difference in the densities between the gelatin/CMC mixture and gelatin-50 when frozen. On the other hand, when we used gelatin-85 to embed the larva in gelatin-50 mold, there was no separation between larva location and gelatin location (Figure 3b).

Furthermore, we noticed that the gelatin-85 dry layer is water soluble, and the gelatin-85 layer surrounding the larvae was removed during washing steps of the H&E staining process. This is advantageous because the resulting H&E sections had minimal interference at the location of larvae as shown in Figure 4a, while gelatin-50 remained on the slide but away from larvae causing no interference. As a comparison, the H&E section of a zebrafish larva embedded in 10% gelatin/5% CMC gives a strong blue background stained by hematoxylin (Figure 4b). This phenomenon was observed for adult zebrafish sectioning in a previous report as well [8]. 

The 3D-printed zebrafish-larvae-shaped mold enables the good control of larvae placement and positioning. After placing the larvae in the mold, the water surrounding the larvae was removed and replaced with embedding medium. At this step, we originally used the 10% gelatin/5% CMC mixture for embedding due to its good sectioning properties, low ion suppression, and preservation of histology of the section [8]. However, as mentioned in Figure 3, the layer separation of gelatin and gelatin/CMC resulted in the failure of this approach. At this point, we hypothesized that a liquid form of gelatin is better than the gelatin/CMC mixture. We discovered that gelatin prepared under 85 °C stays as a liquid at room temperature and embedding with gelatin-50 and gelatin-85 accomplishes minimal layer separation. We believe that gelatin-85 has a greater degree of denaturation than gelatin-50. Although the nature of gelatin prepared under 85 °C was not reported previously according to our literature search, it was reported that gelatin prepared at a higher temperature has a much weaker strength than the gelatin prepared at a lower temperature [15,16]. Basically, gelatin-85 is a “never set” solution at room temperature and this feature is what our embedding process utilized.

#### 2.2.3. The PTFE Surface for Section Placement to Reduce Tissue Folding

We utilized the PTFE surface for section placement on the ITO slide to minimize the larvae section folding during sectioning. After the section was placed on this “non-stick” surface, an ITO slide was placed on the top of the section, a gentle and even pressure was applied to the ITO slide to flatten the section and maximize the contact of the whole section to the surface of a slide. Then, during thawing of the section, different locations of tissue had minimal movement in the horizontal direction resulting in minimal tissue folding and cracking.

### 2.3. Mass Spectrometry Imaging of Zebrafish Larvae

Zebrafish larvae sections were imaged in the positive ion mode with the 2,5-DHA matrix. The coating was applied using sublimation followed by a rehydration or recrystallization step to enhance the signal [17,18,19]. Distinct distributions of ions from endogenous compounds in both the eyes and brain were observed. A selected panel of ions was tentatively identified and named through LipidMaps based on their accurate m/z and the database [20,21]. Specifically, *m*/*z* 703.5601 (cholesterol esters, CE 18:2;O2, [M+Na]^+^) was found in the brain, *m*/*z* 808.5571 (sulfatides, SHexCer 36:0;O3, [M+H-H_2_O]^+^) was found in both the brain and eye, and *m*/*z* 1248.9958 (glycerophosphocholines, PC O-68:7 [M+K]^+^) was found in the retinal area. (Figure 5a). The whole-body ion images and the corresponding average spectrum of the same larva section are presented in the Appendix A.

Zebrafish larvae dosed with clozapine were imaged and shown in Figure 5b (mass spectra from MALDI and DESI are displayed in Appendix A, respectively) and serve as an example of our application of a toxicity study model using zebrafish larvae. Clozapine is an atypical antipsychotic [22], and its effect on zebrafish larvae was reported [23], hence, it was chosen in the proof-of-concept experiment. The MALDI MS images demonstrated the abundance of clozapine in the brain area and body. The imaging results from MALDI and DESI are consistent in that clozapine was found in the upper body of the two separate zebrafish larvae. It worth pointing out that DESI was carried out on an intact larva without sectioning, so it has the advantage of minimal sample preparation and serves as a quick screening and validation tool in the workflow. The correlation of the spatial distribution of clozapine and its effect on zebrafish larvae behavior and the molecular profile is ongoing and is beyond the scope of this report, hence, it will not be discussed further here. 

Mass spectrometry imaging has advanced greatly since its beginning and became an analytical technique both as a complement and replacement to other imaging methods [24]. For sample preparation, tissue treatments for different type of analytes were introduced with a focus on lipids [17,25], small molecules [26], or proteins [18]. For matrix coating, different types of automatic matrix sprayers and sublimators were developed [27]. However, with the rapid expansion of mass spectrometry imaging technology, there is an urgent need for improvements in reproducibility at the sample preparation level [24]. Especially, the step of obtaining consistent sections that is representative of the target organ is rarely focused on. For the application of mass spectrometry imaging in the pharmaceutical environment, it is critical to have sections that are comparable between dosed and non-dosed samples, diseased and normal samples, or among different stages of disease samples to answer the questions, either the drug target engagement, toxicity, or biomarker discovery. Therefore, we focused much effort on optimizing the cryosection for zebrafish larvae. This method can be applied to other small-sized tissues as well. For example, both mouse eyes and spinal cords have diameters of 2–3 mm and will require an embedding process for sectioning. Custom-designed molds for these organs can be used to control the cutting plane and orientation that is critical for downstream analysis and interpretation of the data. 

Mass spectrometry imaging provides land markers of zebrafish larvae and allows virtual dissection of the zebrafish larva to compare tissue-specific drug distribution between molecules. Essentially, mass spectrometry imaging is being used to build an ion atlas or a molecular atlas for zebrafish larvae. This will be a great tool for future applications to select compounds excluded for toxicity target organs and could increase the probability of success (e.g., excluding compounds from the eye in cases of ocular toxicity). Further development could identify organ-specific signatures for easy image segmentation.

## 3. Materials and Methods

### 3.1. Materials

Gelatin from porcine skin (G1890), methanol, ethanol, xylene, acetonitrile, 2,5-dihydroxyacetophenone (DHA), trifluoroacetic acid (TFA), (Hydroxypropyl)-methylcellulose (HPMC, H8384), polyvinylpyrrolidone (PVP, PVP360), and hematoxylin (H3136) were from Sigma/Aldrich (St. Louis, MO, USA). Eosin-Y solution (314–630) was from FisherSci (Pittsburgh, PA, USA). 2, 5-dihydroxybenzoic acid (DHB, D0569) was from Tokyo Chemical Industry (TCI America, Portland, OR, USA) and was recrystallized twice with water. Carboxymethylcellulose sodium (CA192) was from Spectrum Chemical (New Brunswick, NJ, USA). Indium tin oxide (ITO)-coated microscope slides (CG-81IN-S115) were from Delta Technologies (Loveland, CO, USA). The optimal cutting temperature compound (O.C.T., Tissue-Tek, 4583) was from VWR (Radnor, PA, USA).

### 3.2. Sample Preparation

Gelatin-50: 40 g of gelatin was placed in a 250 mL bottle and water (Sigma, 34877) was added to reach the 200 mL volume mark. A spatula was used to mix the gelatin powder and water to minimize the large portions of solid. The cap was placed on the bottle but loosely. The bottle was placed in an oven and the temperature was set at 50 °C overnight. A clear solution of gelatin was obtained. In total, 4 mL of 5% NaN_3_ solution (Ricca, 7144.8-16) was added to prevent bacterial growth. Additional water was added to the bottle to reach a 200 mL volume. This solution was kept under 37 °C before use.

Gelatin-85: 40 g of gelatin was placed in a 250 mL bottle and water (Sigma, 34877) was added to reach the 200 mL volume mark. A spatula was used to mix the gelatin powder and water to minimize the large portions of solid. The cap was placed on the bottle but loosely. The bottle was placed in an oven and the temperature was set at 85 °C overnight. A clear solution of gelatin was obtained. In total, 4 mL of 5% NaN_3_ solution (Ricca, 7144.8-16) was added to prevent bacterial growth. More water was added to the bottle to reach a 200 mL volume. This solution was kept at room temperature before use.

### 3.3. Generating Molds

A 3D printer was used to make a mold form to the specifications of the radial mold form as described in Copper et al. [14]. Gelatin-50, heated to 37 °C, was poured into a small Petri dish, with a diameter of 60 mm and the mold form placed on top of the molten gelatin so that the slot-forming tabs would sink into the gelatin. It was then moved to 4 °C to help ensure proper solidification of the gelatin before attempting to remove the mold form. After about 2 h at 4 °C, the mold form was carefully removed from the mold, which was then ready to receive the larvae.

### 3.4. Animal Husbandry

Adult zebrafish were housed in a Tecniplast ZebTEC stand-alone rack with Active Blue technology. Water parameters were maintained and monitored independently of the housing system capabilities, as follows: pH 7.5 (±0.5), temperature 28 (± 1) °C, and conductivity 500 μS. Husbandry and breeding methods followed those established by Westerfield [28]. Embryos were collected from the breeding tanks within 4 h of removing the barrier between males and females and were allowed to develop and grow in 10 cm Petri dishes (about 50–100 embryos per dish) containing 50 mL water containing 60 μg/mL Instant Ocean Sea salts (Blacksburg, VA). These dishes were kept in an incubator with a 14:10 light dark cycle at 28 °C for 7 days. All experiments were conducted in compliance with AbbVie’s Institutional Animal Care and Use Committee (IACUC) and approved with Protocol # 2207J00003. Date of Approval: 4 August 2022. AbbVie operates under the National Institutes of Health Guide for Care and Use of Laboratory Animals [29] in a facility accredited by the Association for the Assessment and Accreditation of Laboratory Animal Care (AAALAC). No animal health concerns were observed in these studies.

### 3.5. Embedding Larval Zebrafish into Molds

On day 7 post fertilization, the larvae were exposed to clozapine by dissolving the compound in their swimming water using 1% dimethyl sulfoxide as the vehicle. The exposure took place in the incubator at 28 °C with the lights on for 30 min. After the exposure, the larvae were rinsed twice serially by moving them into a set of two Petri dishes containing clean water (about 50 mL each) and swirling them around briefly (10–20 s). The larvae were then moved to a Petri dish containing water containing 0.1 mg/mL Tricaine-S^®^ (Western Chemical, Inc., Ferndale, WA, USA) to anesthetize them. They were then placed into the slots of the mold, one larva per slot, oriented with the head on the widened side of the slot. Once in place, the water was removed from around and beneath the larvae using a fine-tipped pipette and replaced with gelatin-85. A fine-tipped pipette was often needed to remove an air bubble that was trapped below the head after addition of gelatin-85 to the slot. Once six larvae were positioned and covered with gelatin-85, the mold (in the Petri dish) was placed on the surface of finely crushed dry ice. After about 2 min more gelatin-85 was added so that the top surface was flat. This 2 min period allowed for the gelatin-85 surrounding the larvae to begin to solidify and was necessary so that the addition of the supplemental gelatin-85 did not move the larvae out of their slots. The mold was then allowed to fully freeze on the dry ice for about 30 min before it was moved to a −80 °C freezer for additional long-term storage.

### 3.6. Zebrafish Larvae Sectioning Assisted with PTFE Surface Placement

Zebrafish larvae blocks were carefully removed from the housing Petri dish, trimmed, and sectioned on a Thermo HM550 cryostat at −16 °C. The section thickness was set at 12 µm. After the section came off the blade, it was placed on a PTFE microscope slide (Electronic microscope science, 63414-13), then the conductive side of a pre-chilled ITO slide was placed on top of the section, gentle pressure was applied so that the section was flattened and stuck on the ITO slide, and then the ITO slide was separated from the PTFE slide. After this step, sections were thawed with a finger till dryness in the cryostat chamber.

### 3.7. Matrix Coating

Coating by sublimation (for imaging lipids) with a home-built sublimator (Chemglass, VU-0904-281MS): 100 mg of 2,5-DHA was sublimated under vacuum of 12 mTorr on a sand bath of 110 °C for 2 min. Rehydration (recrystallization) of sublimation-coated sections to improve the sensitivity using modified conditions from a previous report [17,18]: two stainless steel plates were attached to the top part of a Petri dish using double-sided thermal conductive tape to form a heat sink, and the slide was placed on the inside of the top part of a Petri dish (FisherSci, FB087579B) with tape. A round filter paper (7 × 7 cm) was placed on the bottom part of the Petri dish and 500 µL of 50% TFA was pipetted on the filter paper evenly. After assembling the top and bottom of the Petri dish and sealing the set with PetriSeal tape (VWR, 490006-878), the Petri dish was placed in the oven under 37 °C for 2 min. The slide was taken out to dry and ready for mass spectrometry acquisition.

The coating for imaging clozapine was DHB matrix by sublimation. The procedure is the same as for DHA, only the duration for sublimation is 3 min. The recrystallization is the same as for DHA as well.

### 3.8. Mass Spectrometry Imaging

MALDI mass spectrometry imaging of matrix-coated sections was carried out on a Bruker timsTOF Flex MALDI2 mass spectrometer. In general, the laser shot was set at 50–100 shots/pixel, and raster step was from 10 µm, trapped ion mobility was active, and theMALDI2 laser was activated for lipid imaging. Data were processed with Bruker SCiLs Lab MVS 2022b Pro.

DESI mass spectrometry imaging was carried out on thaw-mounted whole zebrafish larvae to slides with a ThermoFisher Scientific Q Exactive™ HF Hybrid Quadrupole-Orbitrap™ MS fitted with a Prosolia DESI 2D ionization source. The data were acquired in start point constant velocity mode with a stage of 250 μm s^−1^, yielding a spatial resolution of 50 μm. The N_2_ pressure was 125 PSI, and solvent flow rates were 3.0 μL min^−1^ using the spray solvent mixture of 3/1 (*v*/*v*) methanol/chloroform. The spray voltage was 4.0 kV, the capillary was heated to 300 °C, and the S-lens RF value was 50.0. The Q Exactive™ MS was operated in full scan mode, 250–800 Da, with a resolution of 60,000 and a duty cycle of ~0.200 s. The AGC was set to 5e6 and a maximum injection time of 50 ms, yielding approximately square pixels.

### 3.9. H&E Staining

After mass spectrometry imaging, the slides were immersed in 95% ethanol to remove the matrix coating and stained with H&E [30]. The resulting sections were scanned on a MoticEasyScan Pro 1 slide scanner (Motic Digital Pathology) and images were exported using QuPath [31].

## 4. Conclusions

Sectioning of zebrafish larvae is challenging due to the small size of the larvae, the requirement for organ coverage, and the high throughput for a large sample volume. We developed a method for sectioning zebrafish larvae at high throughput with high reproducibility for desired locations. Gelatin-50 and gelatin-85 are both easy to prepare and use. Embedding larvae with gelatin-85 in gelatin-50 mold results in good quality larvae sections with minimal frozen artifacts, and interference with histological stains and mass spectrometry.

## Figures and Tables

**Figure 1 pharmaceuticals-15-01230-f001:**
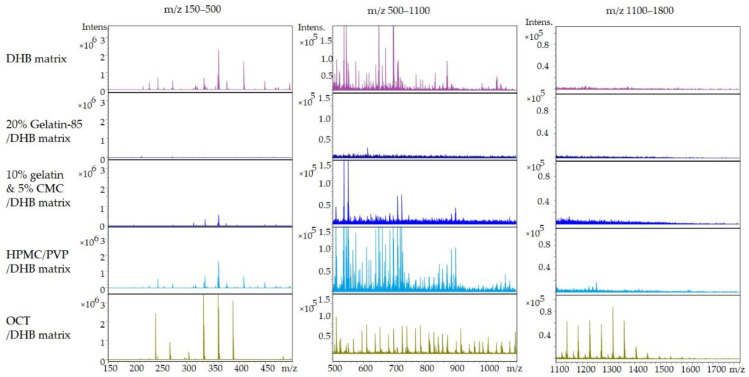
Comparison of the mass spectra of different embedding media under MALDI conditions with DHB matrix coating at three *m*/*z* windows: *m*/*z* 150–500; *m*/*z* 500–1100; *m*/*z* 1100–1800.

**Figure 2 pharmaceuticals-15-01230-f002:**
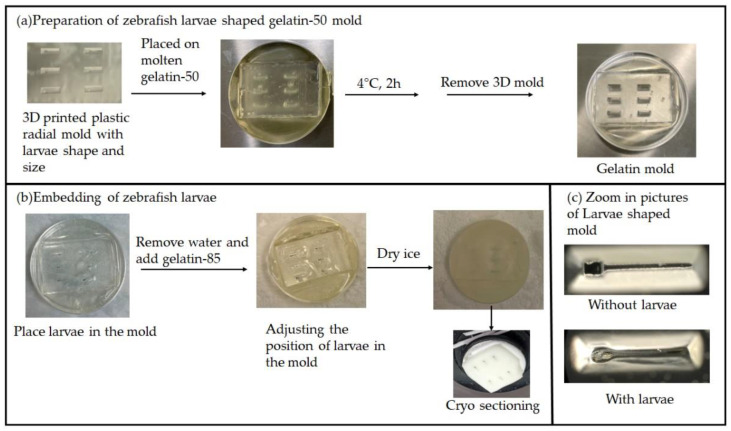
(**a**) Preparation of larvae-shaped gelatin-50 mold with 3D-printed plastic mold; (**b**) embedding of larvae in gelatin using gelatin-50 mold and gelatin-85; (**c**) zoom in pictures of larvae-shaped mold with/without larvae.

**Figure 3 pharmaceuticals-15-01230-f003:**
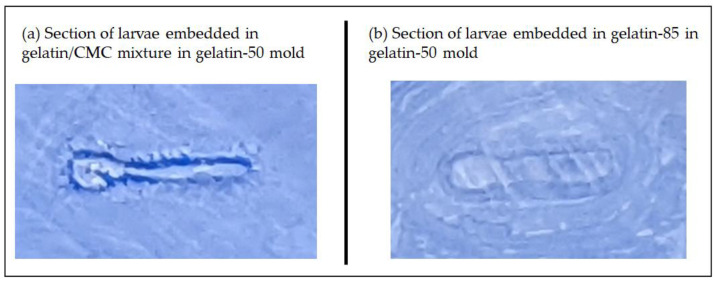
(**a**) Section of larvae embedded in gelatin/CMC then in gelatin-50 mold showing the detachment of gelatin/CMC from gelatin; (**b**) section of larvae embedded in gelatin-85 then in gelatin-50 mold showing perfect embedding of larvae in the gelatin.

**Figure 4 pharmaceuticals-15-01230-f004:**
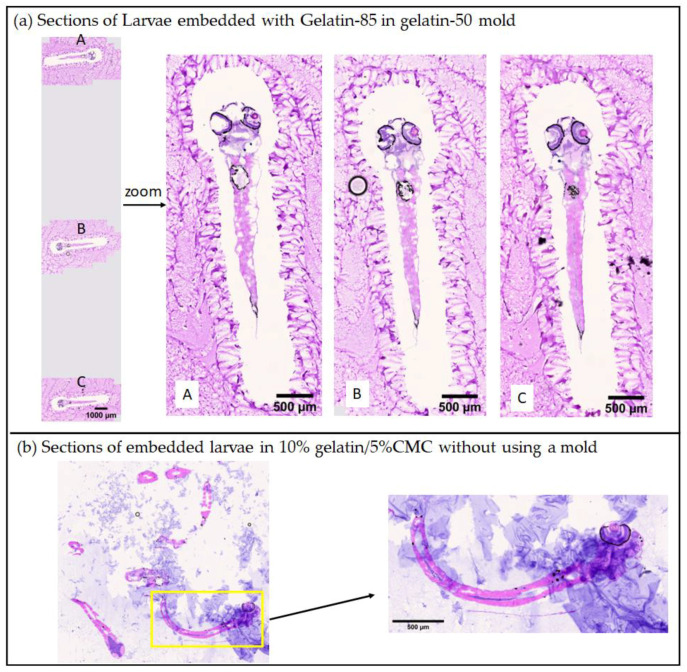
(**a**) Larvae embedded with gelatin-85 in a gelatin-50 mold, resulting in consistent location and plane/angle. The zoom in picture demonstrates that there was no interference from gelatin at the location of larva. A, B, and C were sections from three different larvae in the same block; (**b**) section of embedded larvae in 10% gelatin/5% CMC without using a mold, resulting in randomness of larvae location and section plane/angle, blue background interference within the larva section was observed.

**Figure 5 pharmaceuticals-15-01230-f005:**
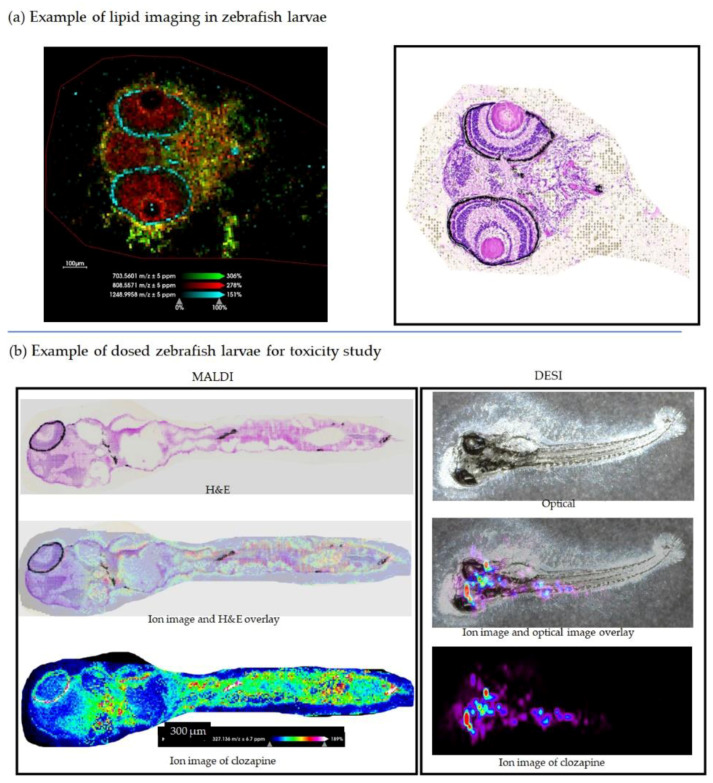
(**a**) Example of three ions distributed on the eyes and brain of zebrafish larvae (left, 10 µm spatial resolution) and H&E-stained optical image (right); (**b**) ion images of clozapine in dosed zebrafish larvae from MALDI (left, 10 µm spatial resolution) and DESI (right, 50 µm spatial resolution) experiments.

## Data Availability

Data is contained within the article and Appendix A.

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
