# Peer review of "Optimization of Zebrafish Larvae Sectioning for Mass Spectrometry Imaging"

_pharmaceuticals, 2022, doi:10.3390/ph15101230_

Round 1

Reviewer 1 Report

The manuscript by Yang et al., details their optimization of fresh-frozen larval zebrafish slices for mass spectrometry imaging. The topic of this manuscript might be interesting, but the organization, text structure and figures in this manuscript are not friendly for readers to read. And there is a serious lack of literatures. In my views, as it stands the data presented are of limited interest and needs to be significantly improved.

Detailed comments:

1. Line 11, “zebrafish” should be capitalized with the first letter.

2. Line 14, the embedding agents chosen by the authors were prepared at 50 and 85°C, whereas the articles reported by Lee et al. and Funke et al. (Anal. Bioanal. Chem., 2021, 413, 2675-2682; Anal. Chim. Acta, 2021, 1177, 338770) revealed that the preparation temperature of gelatin does not exceed 60 ℃, why did the author choose gelatin prepared at 85 ℃ for subsequent experiments?

3. Lines 18-22, besides, some sentence tenses in this article are confused, please check the full text for correction.

4. Line 25, the format of introduction is recommended to be consistent with that of the whole article.

5. Line 34, the authors demonstrated that zebrafish embryos can develop major mammalian organ systems as early as 7 days.  Please add literature.

6. Line 51, with respect to zebrafish larvae, please also consider the recent literature (Anal. Bioanal. Chem., 2021, 413, 5135-5146).

7. Line 76, “mass spec imaging” should be changed to “mass spectrometry imaging”. There are still some errors in the text, such as Line 151 and Line 269, etc. Please check the full text for revision. 

8. Figure 1, the resolution is too low to acquire any spectral information (e.g., peak annotation, X- and Y-coordinates), please replace them. And “0.00” intensity in some graphs should be unified or kept at the same horizontal line. “150-500 m/z; 500-1100 m/z; 1100-1800 m/z” should be revised to “m/z 150-500; m/z 500-1100; m/z 1100-1800”.

9. Lines 89-93, beyond [10, 11], other recent references about sample preparation optimization of zebrafish for MSI could have been added here. Such as Anal. Bioanal. Chem., 2022, 414, 4777-4790.

10. Line 101, the layout of Figure 2 is suggested to be rearranged, which does not give a good feeling (for example, the arrows are not neatly placed). In addition, there is no relevant description of Figure 2 in the main text.

11. Line 115, the distance between the text and the image in Figure 3b is suggested to be rearranged.

12. Lines 129-137, the results described by the authors are more like illustrating Figure 2 than Figure 4. Additionally, the reviewers suggest the content of section 2.2.2 should be described in advance and then section 2.2.1 for the advantages of different gelatin solutions.

13. Figure 4a, the two zebrafish larvae are not the same, please comment it. And the scale bars are too small to observe in Figure 4a-b. The arrow is suggested to change to box marker for enlarging.

14. Lines 172-173, materials and methods are repeated and singular and plural are not uniform.

15. Line 290, it would be nice if the authors provided more data to verify the reproducibility of the method.

16. In general, the largest surface of zebrafish is selected for slicing, and the author chooses the direction from top to bottom for slicing. What is the reason? 

17. Please add the full names of CMC.

18. In the legend of the article, please pay attention to the case of the first letter and keep it consistent. 

19. Conclusions, please supplement more discussion/conclusions and insightful perspective.

Author Response

The manuscript by Yang et al., details their optimization of fresh-frozen larval zebrafish slices for mass spectrometry imaging. The topic of this manuscript might be interesting, but the organization, text structure and figures in this manuscript are not friendly for readers to read. And there is a serious lack of literatures. In my views, as it stands the data presented are of limited interest and needs to be significantly improved.

Detailed comments:

  1. Line 11, “zebrafish” should be capitalized with the first letter.

Changed

  1. Line 14, the embedding agents chosen by the authors were prepared at 50 and 85°C, whereas the articles reported by Lee et al. and Funke et al. (Anal. Bioanal. Chem.2021413, 2675-2682; Anal. Chim. Acta20211177, 338770) revealed that the preparation temperature of gelatin does not exceed 60 ℃, why did the author choose gelatin prepared at 85 ℃ for subsequent experiments?

We do agree with reviewer that reasoning was not very clear why gelatin was prepared under 85°C, so we added following sentence and literature in the discussion:

“We discovered that gelatin prepared under 85°C stays as a liquid at room temperature and embedding with gelatin-50 and gelatin-85 accomplishes minimal layer separation. We believe gelatin-85 has greater degree of denaturation more than gelatin-50. Although the nature of gelatin prepared under 85°C was not reported before with our literature search, it was reported that gelatin at higher temperature has much weaker strength than the gelatin at lower temperature does [22, 23]. Basically, gelatin-85 is a “never set” solution at room temperature and this feature is what embedding needs.”

  1. Lines 18-22, besides, some sentence tenses in this article are confused, please check the full text for correction.

Line 18-19, Sentence “During sectioning, PTFE surface was used to assist section placement. Using these conditions, the larvae sections with minimum folding and defect were obtained.” was removed.

Line 20, “from gelatin-85” was removed.

  1. Line 25, the format of introduction is recommended to be consistent with that of the whole article.

Changed the format accordingly.

  1. Line 34, the authors demonstrated that zebrafish embryos can develop major mammalian organ systems as early as 7 days.  Please add literature.

Literature added: Strahle, U.;  Scholz, S.;  Geisler, R.;  Greiner, P.;  Hollert, H.;  Rastegar, S.;  Schumacher, A.;  Selderslaghs, I.;  Weiss, C.;  Witters, H.; Braunbeck, T., Zebrafish embryos as an alternative to animal experiments-A commentary on the definition of the onset of protected life stages in animal welfare regulations. Reprod Toxicol 2012, 33 (2), 128-132.

  1. Line 51, with respect to zebrafish larvae, please also consider the recent literature (Anal. Bioanal. Chem.2021413, 5135-5146).

This Literature was added and the reason why we need to develop new approach of embedding procedure was discussed.

  1. Line 76, “mass spec imaging” should be changed to “mass spectrometry imaging”. There are still some errors in the text, such as Line 151 and Line 269, etc. Please check the full text for revision. 

All places with “mass spec imaging” are changed to “mass spectrometry imaging”

  1. Figure 1, the resolution is too low to acquire any spectral information (e.g., peak annotation, X- and Y-coordinates), please replace them. And “0.00” intensity in some graphs should be unified or kept at the same horizontal line. “150-500 m/z; 500-1100 m/z; 1100-1800 m/z” should be revised to “m/z 150-500; m/z 500-1100; m/z 1100-1800”.

Figure 1 is changed accordingly with the font size, coordinate markers, and “m/z” is moved to the front of numbers. A different scaling of each mass window was also added in supplementary (Figure S1), where the scaling was to the highest peak of each mass window.

  1. Lines 89-93, beyond [10, 11], other recent references about sample preparation optimization of zebrafish for MSI could have been added here. Such as Anal. Bioanal. Chem.2022414, 4777-4790.

Literature was added accordingly

  1. Line 101, the layout of Figure 2 is suggested to be rearranged, which does not give a good feeling (for example, the arrows are not neatly placed). In addition, there is no relevant description of Figure 2 in the main text.

The layout of Figure 2 was adjusted, arrows were changed with larger size and black color. Description of Figure 2 was added. The image of cutting block from Figure 3 was moved to Figure 2.

  1. Line 115, the distance between the text and the image in Figure 3b is suggested to be rearranged.

Figure 3 was totally modified with the removal of FFTE slide application and cutting block to streamline the talking points.

  1. Lines 129-137, the results described by the authors are more like illustrating Figure 2 than Figure 4. Additionally, the reviewers suggest the content of section 2.2.2 should be described in advance and then section 2.2.1 for the advantages of different gelatin solutions.

Yes, the original text was confusing, and it is now edited as suggested.

  1. Figure 4a, the two zebrafish larvae are not the same, please comment it. And the scale bars are too small to observe in Figure 4a-b. The arrow is suggested to change to box marker for enlarging.

Figure 4 was modified: 4a) image of array of three larvae section was added, zoom in picture of each larva was added. 4b) zoom in picture of the boxed larva was replaced with the corresponding picture.

  1. Lines 172-173, materials and methods are repeated and singular and plural are not uniform.

“materials and method” at line 173 was removed.

  1. Line 290, it would be nice if the authors provided more data to verify the reproducibility of the method.

Example of larvae in array mold was added in Figure 4a)

  1. In general, the largest surface of zebrafish is selected for slicing, and the author chooses the direction from top to bottom for slicing. What is the reason? 

In our routine application, the zebrafish was sectioned from top to bottom, all sections (~30) were collected and analyzed with mass spectrometry imaging or optical imaging. Examples in Figure 4 were only for demonstration purpose. 

  1. Please add the full names of CMC.

The full name of CMC is added at the location where it appears the first case (now is in line 53)

  1. In the legend of the article, please pay attention to the case of the first letter and keep it consistent. 

The letter cases in the legends are adjusted now.

  1. Conclusions, please supplement more discussion/conclusions and insightful perspective.

More discussions are added.

Reviewer 2 Report

Overall an interesting paper, but lack a lot in details and explanations. This si not on the quality level that I would expect from a scintific paper its more like a student report. This has to be imporved when it comes to explaining and showing data in a well explained and detail manner. 

These are some major points that need to be adressed:

l 81 -82and figure 1

Among the embedding media that we tested, gelatin-85 and gelatin/CMC mixture pro vide minimal mass spectrum background hence little interference for the detection of small molecules (200-1200 m/z), as shown in Figure 1.

This statement is vague and the data not described well enough to confirm this statement. The quality of figure 1 is also very bad and you can't discern individual peaks. This has to be improved and better described.

L152

We imaged lipid range species in the zebrafish larvae section focusing on the head

Which lipids, you have some odd abbreviation in the figure, but they are not explained. where are the mass spectra?

L 158

MALDI MS images demonstrated the abundance of clozapine in the brain area and body.

What peak did you image here, why did you image that peak, where is the mas spectra

L 159

The imaging results from MALDI and DESI has good agreement.

How are they in good agreement, you just state this but you show no real correlation

Author Response

Overall, an interesting paper, but lack a lot in details and explanations. This is not on the quality level that I would expect from a scientific paper its more like a student report. This has to be improved when it comes to explaining and showing data in a well explained and detail manner. 

These are some major points that need to be addressed:

l 81 -82and figure 1

“Among the embedding media that we tested, gelatin-85 and gelatin/CMC mixture provide minimal mass spectrum background hence little interference for the detection of small molecules (200-1200 m/z), as shown in Figure 1.”

 This statement is vague, and the data not described well enough to confirm this statement. The quality of figure 1 is also very bad and you can't discern individual peaks. This has to be improved and better described.

 Figure 1 has been edited with bigger fonts, and reformatted legends, X and Y coordinate markers. A different scaling of each mass window was also added in supplementary (Figure S1), where the scaling was to the highest peak of each mass window.

Further description of the mass spectra of different media was added.

L152

“We imaged lipid range species in the zebrafish larvae section focusing on the head”

Which lipids, you have some odd abbreviation in the figure, but they are not explained. where are the mass spectra?

Mass spectra and more ion images of this lipid imaging session was added in the supplementary. The lipid identification (tentative) was described with citations to the LipidMaps.

 L 158

“MALDI MS images demonstrated the abundance of clozapine in the brain area and body.”

What peak did you image here, why did you image that peak, where is the mass spectra

The reason why we chosen clozapine as dosing compound was added. Mass spectra of this imaging session was added in the supplementary Figure S3 and S4.

 L 159

“The imaging results from MALDI and DESI has good agreement.”

How are they in good agreement, you just state this but you show no real correlation

More description is added in this regard.

Round 2

Reviewer 1 Report

The authors have carefully addressed my concerns. Considering that the quality of this revised manuscript has been improved, it can be accepted for publication as it is.

Reviewer 2 Report

this version is much improved. No further comments.